# Influence of Parturition on Rumen Bacteria and SCFAs in Holstein Cows Based on 16S rRNA Sequencing and Targeted Metabolomics

**DOI:** 10.3390/ani13050782

**Published:** 2023-02-21

**Authors:** Yansheng Guo, Feifei Wang, Yongxia Mao, Weiyi Kong, Jiandong Wang, Guijie Zhang

**Affiliations:** 1College of Agriculture, Ningxia University, Yinchuan 750021, China; 2Institute of Animal Sciences, Ningxia Academy of Agricultural and Forestry Sciences, Yinchuan 750002, China

**Keywords:** parturition, dairy cows, rumen bacterial communities, short-chain fatty acids

## Abstract

**Simple Summary:**

The depressed appetite at parturition might induce the changes in the composition and quantity of rumen microbiota. At present, little is known about the influence of parturition on the levels of rumen microbiota and their fermentation ability in dairy cows. The objective of this study was to evaluate the effects of parturition on composition and quantity of bacterial communities and concentrations of short-chain fatty acids (SCFAs) with 16S rRNA high-throughput sequencing and targeted GC–MS/MS metabolomics. The results showed that parturition altered the levels of rumen bacteria and their fermentation ability. The findings provide a better understanding of the effect of parturition on rumen digestive function.

**Abstract:**

The rumen fluids from ten cows at Day 3~5 before calving and Day 0 after calving were collected to analyze the composition and quantity of bacterial communities and concentrations of SCFAs. The results showed that the relative abundances of unidentified *Lachnospiraceae, Acetitomaculum, Methanobrevibacter, Olsenella, Syntrophococcus, Lachnospira,* and *Lactobacillus* genera were significant increased (*p* < 0.05), while that of unidentified*-Prevotellaceae* was notably decreased after calving (*p* < 0.05). In addition, the concentrations of acetic acid, propionic acid, butyric acid, and caproic acid obviously decreased after calving (*p* < 0.01). Our findings show that parturition altered the rumen microbiota and their fermentation ability in dairy cows. This study defines a rumen bacteria and metabolic profile of SCFAs associated with parturition in dairy cows.

## 1. Introduction

Parturition is a complex physiological process modulated by the endocrine, nervous, and immune systems, and other factors, during which the metabolism of hormones, sugars, proteins, and lipids is severely disturbed. Among these factors, alterations in hormones play a crucial role in the loss of appetite during late gestation in dairy cows [1,2]. On Days 3~5 before parturition, DMI is reduced sharply so that the daily feed intake is only approximately 8~9 kg per dairy cow [3]. Due to the limited feed intake, rumination time has been observed to be largely decreased about 8 h before delivery and gradually recovered about 6 h later [4]. Hence, rumination time has been well recognized as a means of detecting parturition in dairy cows [5].

Rumen microbiota, consisting of anaerobic bacteria, archaea, protozoa and fungi, ferment fibrous and nonfibrous sources of carbohydrates in the feed into SCFAs to supply energy [6]. Rumination is universally acknowledged as the primary means by which the size of feed particles consumed is decreased in dairy cattle [4]. More sites for microbial attachment are exposed when the feed particle is ruminated. During rumination, saliva is added and the accumulated CO_2_ and VFA are released via chewing, which can make the microenvironment more beneficial to bacterial growth [7]. Although the importance of rumination for microbial activities by reducing particle size is well recognized, the detailed effect of parturition on the quantity and fermentation activity of rumen microbial communities are not well reported and require further study.

High-throughput sequencing is a category of powerful technologies for obtaining high-coverage information on the classification and diversity of microbial communities without isolation and culture [8], among which 16S rRNA sequencing has been commonly applied to explore the composition of microbial communities in the rumens and intestines of dairy cows [9,10]. Targeted GC–MS/MS metabolomics are suitable for identifying and quantifying small molecule acids, amino acids, sugars, and fatty acids, and have also been successfully used to check the concentrations of these metabolites in the serum, feces, and rumen fluid of ruminants [11,12,13]. Consequently, we collected the rumen fluids from ten cows on Days 3~5 before calving and Day 0 after calving to reveal the influence of parturition on rumen bacterial communities and SCFAs using 16S rRNA high-throughput sequencing and targeted GC–MS/MS metabolomics, attempting to provide more information on physiological process of parturition in Holstein dairy cows.

## 2. Materials and Methods

### 2.1. Ethical Statement

The experimental scheme was approved by the Animal Ethics Committee of Ningxia University (authorization number: 025/22) and complied with the international guidelines for animal experiments.

### 2.2. Sample Collection

At a large modern enterprise around the Yinchuan city of China, twenty healthy Holstein cows on Days 3~5 before calving (2~3 parity and 3.2~3.5 BCS) were selected to collect rumen fluids. The cows were fed three times each day with TMR diet (Appendix A) and DMI was 9.2 ± 0.6 kg·d^−1^ during calving. The concentrate-to-forage ratio of TMR diet was 3.6:1. Their body temperatures were normal and blood ketones values were around 1.0. Each cow had taken rumen fluid at 1 h after their second meal. Ultimately, the rumen fluids from ten cows were retained as “samples before parturition” because the calving date of the ten cows coincided with expected calving date. Then, the rumen fluids as “samples after parturition” were taken from the same ten cows at 1 h after calving without eating. Rumen fluids before parturition were labeled as E1~E10 (Group E), and those after parturition were marked as A1~A10 (Group A).

The rumen fluids were taken using special collectors equipped with a metal filter at one end and a 50 mL syringe at the other end. Discarding the first tube of rumen fluid and the second tube of rumen fluid was used as the test sample. The rumen fluids were filtered by four layers of sterilized gauze, transferred to cryopreservation tubes and stored at −80 °C.

### 2.3. Targeted GC–MS/MS Metabolomics Analysis of Rumen SCFAs

The rumen fluids were well mixed in maximum vortex frequency after slowly thawing at 4 °C. Fifty microliters of rumen fluid was vortexed with 100 μL of 36% chromatographic grade phosphoric acid solution for approximately 3 min in the eppendorf tube and then vortexed with 150 μL of chromatographic grade MTBE (methyl tertiary butyl ether) solvent containing internal standard to extract SCFAs from rumen samples with ultrasonication for approximately 5 min in an ice bath. The extracted solution was centrifuged at 12,000× *g* r/min and 4 °C for 10 min, and 90 μL of supernatant was absorbed into the sample bottle with a glass liner for later targeted GC–MS/MS analysis.

An Agilent 7890A–5975C gas chromatograph-mass spectrometer (Agilent Technologies, Santa Clara, CA, USA) with a quadrupole analyzer was used for the qualitative and quantitative analysis of rumen SCFAs. According to previous literature [14], the GC–MS/MS conditions were developed (Appendix A). The high reliability of the GC–MS/MS conditions was confirmed with intraday precisions, interday precisions, and recovery rates of SCFAs with different concentrations of QC samples (Appendix A), and good stability of the instrument was observed through the total ion current [15] overlap of QC sample mass spectrometry (Appendix A). The chromatographic peaks representing acetic acid, propionic acid, isobutyric acid, butyric acid, isovaleric acid, valeric acid, and caproic acid were ensured according to the ion pair information, secondary spectrum and retention time (Appendix A). The linear regression equations of the seven SCFAs were established based on the peak intensities of SCFA standard solutions with serial concentrations (Table 1). The concentrations of SCFAs in rumen fluid were calculated based on linear regression equations (Appendix A). The file in the csv format containing names of SCFAs, concentrations and samples information was imported into MetaboAnalyst 5.0 (https://www.metaboanalyst.ca/, accessed on 8 October 2021) to carry out the following metabolomics analysis. Principal component analysis (PCA) was used to visualize the changes in the metabolic profile of the seven rumen SCFAs before and after parturition. Variable importance in projection (VIP) values of the seven SCFAs were obtained by partial least squares-discriminant analysis (OPLS-DA). The *p* value and fold change (FC) were also acquired by *t* test. SCFAs with VIP ≥ 1, FC ≥ 1.5, and *p* < 0.05 were selected as biomarkers for differentiating dairy cows before and after parturition.

### 2.4. 16S rRNA High-Throughput Sequencing of Rumen Bacteria Communities

The SDS method was used to extract the genomic DNA of rumen bacterial communities, and the purity and concentration of genomic DNA were evaluated with 1% AGE. Next, 341F (CCTAYGGGRBGCASCAG) and 806R (GGACTACNNGGGTATCTAAT) primers with barcodes were designed to amplify the V3~V4 hypervariable regions of the rumen bacterial 16S rRNA gene [16,17]. The PCRs were carried out using a thermal cycle PCR system (Gene Amp 9700, ABI, Waltham, MA, USA) according to the published literature [17]. The amplified products were validated using 2% agarose gel electrophoresis and further purified using the Qiagen gel extraction kit (Qiagen, Hilden, Germany). After amplification, DNA libraries were constructed with a TruSeq DNA PCR-Free Sample Preparation Kit (Illumina, San Diego, CA, USA) and quantified with Qubit and Q-PCR methods. The qualified libraries were subsequently sequenced with a PE250 strategy via the NovaSeq6000 platform (Illumina Inc., San Diego, CA, USA).

The reads of each sample were separated after removing the barcode and the primer sequences and then spliced into raw tags using FLASH (V1.2.7, http://ccb.jhu.edu/software/FLASH/ accessed on 9 January 2023) [18]. The raw tags were quality-filtered to obtain high-quality tags (clean tags) [19]. The clean tags were truncated and filtered via QIIME (V1.9.1, http://qiime.org/scripts/split_libraries_fastq.html accessed on 9 January 2023) [20], and chimeras were further removed with VSEARCH (https://github.com/torognes/vsearch/ accessed on 9 January 2023) [21]. The effective tags of all samples were finally obtained after the above data processing and then clustered into operational taxonomic units (OTUs) at 97% identity using UPARSE (v7.0.1001, http://www.drive5.com/uparse/ accessed on 9 January 2023) [22]. Meanwhile, the representative sequence of OTUs was selected according to the UPARSE algorithm. Species annotation and taxonomic analysis of each 16S rRNA gene sequence was performed with Mothur and the SSUrRNA database of SILVA132 (http://www.arb-silva.de/ accessed on 9 January 2023) with a threshold of 0.8~1 [23]. The bacterial community composition of each sample was assessed at the levels of kingdom, phylum, class, order, family, genus, and species.

Alpha diversity indices of the groups, including Shannon, Simpson, Chao1, and ACE, were calculated with QIIME after homogenizing the data of each sample. The program R (Version 2.15.3) was used to generate the rarefaction curve and *t* test of alpha diversity between groups. Principal coordinate analysis (PCoA) based on unweighted UniFrac distance was selected to visualize beta diversity using WGCNA, stats, and ggplot2 packages in R. Anosim analysis based on Bray–Curtis distance were used to determine the difference in bacterial communities between two groups with the anosim function of the vegan package in R. The *t* test in R was used to search and visualize differential species between the two groups at the genus level. LEfSe (LDA Effect Size) software was used to find biomarkers for the two groups and generate an LDA score cladogram. Spearman correlation analysis between rumen bacterial genera with the top 30 abundances and SCFAs was carried out using MicrobiomeAnalyst 2.0 (https://www.microbiomeanalyst.ca/ accessed on 9 January 2023).

## 3. Results

### 3.1. Changes in the Metabolic Profile and Concentrations of Rumen SCFAs

The 2D scatter plots of PCA and OPLS-DA exhibited a clear separation of the metabolic profiles of the seven SCFAs between Groups E and A (Figure 1a,b), indicating that rumen fermentability in dairy cows obviously changed during parturition. Acetic acid, propionic acid, butyric acid, and caproic acid could be metabolic biomarkers for differentiating the rumen fluids before and after calving according to their VIP values (≥1), FC (≥1.5) *p* values (<0.05), and their concentrations in rumen fluid presented a significant decreasing trend during parturition. The average concentrations, pH of rumen liquids, *p* values, FC, and VIP values are shown in Table 1.

### 3.2. Alteration in Rumen Bacteria Communities

An average of 1,399,432 raw tags were detected by 16S rRNA gene sequencing from the 20 rumen fluid samples of dairy cows before and after parturition, and 837,879 effective tags with an average length of 413 bp were ultimately screened after quality control and filtration (Appendix A). A total of 2590 OTUs were acquired by clustering at 97% identity, of which 1965 OTUs were shared by both groups. The species accumulation curve was prone to be flat when the number of samples were higher than 20, showing that the number of samples in this study was reliable for estimating species richness (Figure 2a). The rarefaction curve of alpha diversity tended to be flat as the sequencing depth increased, indicating that the sequencing results could reflect the diversity of bacterial communities in rumen fluid samples and that there would be no large number of new OTUs to appear even if the sequencing depth was further increased (Figure 2b). The Shannon, Simpson, Chao1, and ACE indices of alpha diversity reflecting the within-group diversity and abundance of the bacterial community are listed in Table 2. The Shannon, Chao1, and ACE indices in Group A were significantly higher than those in Group E (*p* < 0.05). The 2D scatter plot of PCoA by unweighted UniFrac distance of beta diversity showed a visible separation between Groups A and E (Figure 3a). As shown in Figure 3b, the nonparametric test of anosim by the Bray–Curtis distance proved that the difference between groups was significantly greater than that within groups (R = 0.399, *p* = 0.001).

Five kinds of bacterial communities from phylum to species were identified as biomarkers for Groups E and A with LEfSe analysis, including *f_Methanobacteriaceae, o_Methanobacteriales, c_Methanobacteria, f_Prevotellaceae,* and *f_Lachnospiraceae* (Figure 4). Furthermore, genera with the top 30 abundances in the two groups were selected for the *t* test, and eighteen rumen bacterial genera were ascertained to be significantly different in relative abundance between the two groups (Figure 5). Eight major abundant genera (accounting for 0.05% of the total sequences in at least one sample) with differences multiple ≥2 were further confirmed as differential species between the two groups (Table 3). Among these eight genera, the relative abundances of *unidentified_Lachnospiraceae*, *Acetitomaculum*, *Methanobrevibacter*, *Olsenella*, *Syntrophococcus*, *Lachnospira,* and *Lactobacillus* in Group A were significantly higher than those in Group E. In contrast, the relative abundance of *unidentified_Prevotellaceae* in Group A was significantly lower than that in Group E.

### 3.3. Correlation between Rumen Bacteria and SCFAs

Correlation analysis was performed on rumen bacterial genera in group A and group E. The bacterial genera and SCFAs with |r| > 0.4 and *p* ≤ 0.05 were listed in Table 4. The results showed that acetic acid was significantly negatively correlated with *Syntrophococcus*, *Acetitomaculum*, *Lactobacillus*, unidentified_*Enterobacteriaceae*, and *Methanobrevibacter* (*p* < 0.05 or *p* < 0.01). Propionic acid was significantly negatively correlated with *Olsenella*, unidentified_*Lachnospiraceae*, *Desulfobulbus*, unidentified_ *Enterobacteriaceae,* and *Lachnospira* (*p* < 0.05 or *p* < 0.01), and significantly positively correlated with unidentified_ *Prevotellaceae* (*p* < 0.05). Butyric acid was significantly negatively correlated with *Subdoligranulum*, *Lactobacillus*, *Acetitomaculum*, *Syntrophococcus*, *Solobacterium*, *Desulfobulbus,* and unidentified_ *Enterobacteriaceae* (*p* < 0.05 or *p* < 0.01).

## 4. Discussion

The current study investigated the influence of parturition on the phenotype composition and quantity of bacteria in ruminal liquid of dairy cows using 16S sequencing technique. At present, either liquid or solid fractions in rumen are widely used for microbiome analysis. The two fractions have differentiated ecological niches [24], however, the high degree of similarity in microbial community composition, diversity, and relative abundance profiles between the two fractions in cattle and sheep have also been affirmed by a series of studies [25,26,27,28]. Considering this, the ruminal liquid is selected for the current study, which is easy to collect and poorly contaminated.

Ruminal SCFAs, originating from microbial fermentation of feed carbohydrates, are absorbed by the rumen epithelium as a dominant source of energy for ruminants. Acetic acid is the most-produced compound, accounting for approximately 70~75% of the total production of SCFAs and supplying energy through the tricarboxylic acid cycle [29]. Propionic acid generates approximately 50~60% of glucose via hepatic gluconeogenesis [30]. Butyric acid accounts for 10~20% of the total production of SCFAs and is transformed into β-hydroxybutyric acid to provide energy for muscle tissue [31], and it also plays an important role in the regulation of innate and adaptive immune cell generation and function [32]. An intense negative energy balance at parturition is most likely induced by depressed appetite and the initiation of milk synthesis in dairy cows [33]. Although dairy cows can mobilize adipose tissue into fatty acids to remedy the energy deficit, excessive lipolysis heightens the risk for metabolic and inflammatory diseases [34]. The parturition period is critical for determining the potential of the cow mammary gland to synthesize milk, and the degree of that potential depends on how much nutrients are received by the gland [35]. In this study, the concentrations of acetic acid, propionic acid, and butyric acid in the rumen fluid after parturition were significantly lower than those before parturition, indicating that parturition probably aggravated negative energy balance, which in turn negatively affected milk production performance and increased the risk for postpartum metabolic and inflammatory diseases.

Bacteria population acts as a key role in digestive and metabolic activities of rumen, obtaining energy from fiber, starch, sugars, and protein of feed. In our study, alpha diversity and beta diversity of rumen bacteria are distinctly altered before and after calving, indicating rumen bacterial community composition and its digestive and metabolic activities were affected by parturition. The relative abundances of rumen bacteria in genera level was further observed in our study to characterize their change feature during parturition, showing that seven bacteria abundances increased including *Lachnospiraceae*, *Acetitomaculum*, *Methanobrevibacter*, *Olsenella*, *Syntrophococcus*, *Lachnospira* and *Lactobacillus*, and *Prevotellaceae* abundances decreased after calving.

*Prevotellaceae* is the dominant flora, accounting for approximately 60%~70% of rumen microbes of dairy cows, which can decompose the protein and carbohydrates in the feed into propionic acid, lactic acid, and succinic acid [36]. Approximately 90% of glucose in dairy cows is generated by gluconeogenesis, of which 50–60% originates from propionic acid through hepatic gluconeogenesis [37]. In this study, propionic acid shows a positive correlation with *Prevotellaceae* in concentration before and after calving. Numerous studies have shown that dairy cows are prone to negative energy balance after calving [38]. Therefore, the decreasing in *Prevotellaceae* abundance is most likely be one of reasons for the negative energy balance in postpartum dairy cows.

In the rumen, methane production is beneficial to the microbial growth and digestion by regulating the partial pressure of hydrogen [39]. *Methanobrevibacter* can utilize some metabolites, such as acetic acid, propionic acid, and H_2_, to generate methane [40], and a higher levels of methane emissions is correlated with high abundance of *Methanobrevibacter* [41]. In this study, the abundance of *Methanobrevibacter* is lower before than after calving, which is speculated to be related to the inhibition of rumination during parturition.

As the abundance of *Methanobrevibacter* decreases, several acetic acid-producing bacteria (*Lachnospiraceae, Acetitomaculum, Lactobacillus, Olsenella,* Syntrophococcus, and Lachnospira) have declined before calving in this study. Family *Lachnospiraceae* has been verified as the predominant acetogen in the rumen fermentation system of dairy cows [42]. The genus *Acetitomaculum* exhibits a significant positive association with lower feed efficiency [25], which can ferment monosaccharides into acetic acid [43]. *Lactobacillus* can hydrolyze starch and other sugars to produce acetic acid, butyric acid, and lactic acid [44]. *Olsenella* has been found in GIT of human and animals, fermenting starch and glycogen substrates and producing lactic, acetic, and formic acid [15]. *Syntrophococcus* produces acetate only from pyruvate and various carbohydrates although its role is not well-understood in the rumen metabolism [45]. *Lachnospira* can produce pectin lyases and release into extracellular environment to decompose pectin to oligogalacturonides, which are metabolized as acetic acid within the cell [46]. However, we find that the concentrations of acetic acid (including butyrate) in rumen are significantly higher before calving than after calving, which is inconsistent with these bacterial changes. The reason for this inconsistency is most likely related to the large amount of energy required for postpartum lactation initiation in dairy cows.

## 5. Conclusions

Our study investigated the composition of the bacterial communities and the concentrations of SCFAs in the rumen of dairy cows before and after calving with 16S rRNA high-throughput sequencing and targeted GC–MS/MS metabolomics. Distinct alteration in the composition of rumen bacteria before and after calving could attributed to the sharply decreasing feed intake during parturition. As a dominant flora, the decreasing in levels of *Prevotellaceae* of propionic acid-producing bacteria are most likely be one of the reasons for the negative energy balance in postpartum dairy cows. Lower *Methanobrevibacter* abundance before calving may be related to the inhibition of rumination during parturition. On the other hand, the change in abundance of several acetic acid-producing bacteria is inconsistent with that of acetic acid before and after calving, which is most likely related to the large amount of energy required for postpartum lactation initiation in dairy cows. Our current study only focused on rumen bacterial populations; it provided a more comprehensive understanding of parturition on the dairy cow microbiota and as to how scientists would investigate all major microbial populations within therumen, including protozoa and fungi.

## Figures and Tables

**Figure 1 animals-13-00782-f001:**
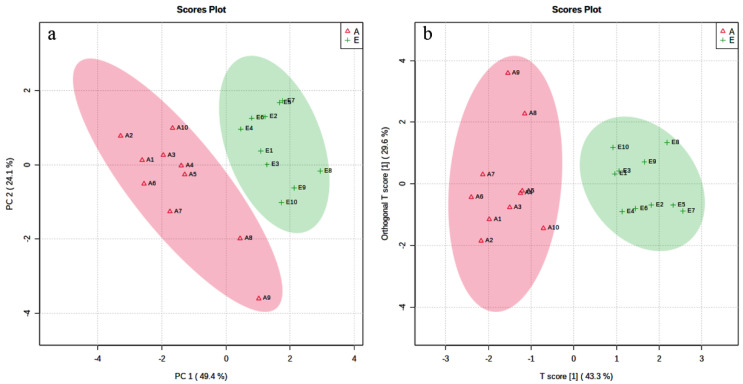
2D scatter plots of principal component analysis (PCA) and partial least squares-discriminant analysis (OPLS−DA) of rumen SCFAs in dairy cows. Note: (**a**): 2D scatter plot of PCA; (**b**): 2D scatter plot of OPLS−DA. E: before parturition; A: after parturition.

**Figure 2 animals-13-00782-f002:**
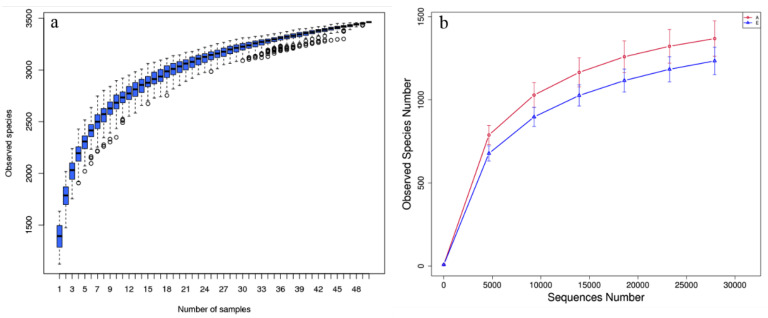
Species accumulation curve (**a**) and rarefaction curve of alpha diversity (**b**).

**Figure 3 animals-13-00782-f003:**
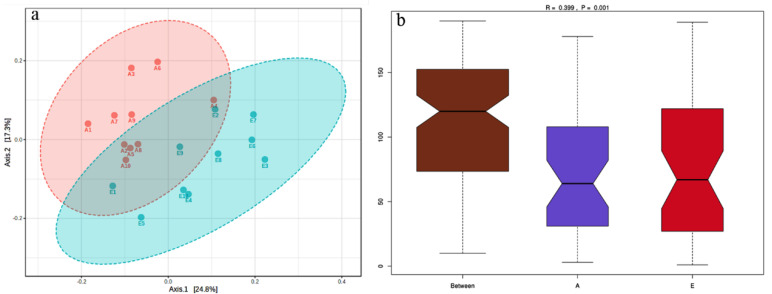
2D scatter plot of principal coordinate analysis (PCoA) based on unweighted UniFrac distance (**a**) and boxplot of ANOSIM based on Bray−Curtis distance (**b**) of rumen bacterial communities. A, after parturition; E, before parturition. The *y*−axis is the order of the distance between the samples. On the *x*−axis, A and E means within Groups A and E. An R−value > 0 indicates a difference between groups greater than that within groups, and *p* < 0.05 indicates a significant difference.

**Figure 4 animals-13-00782-f004:**
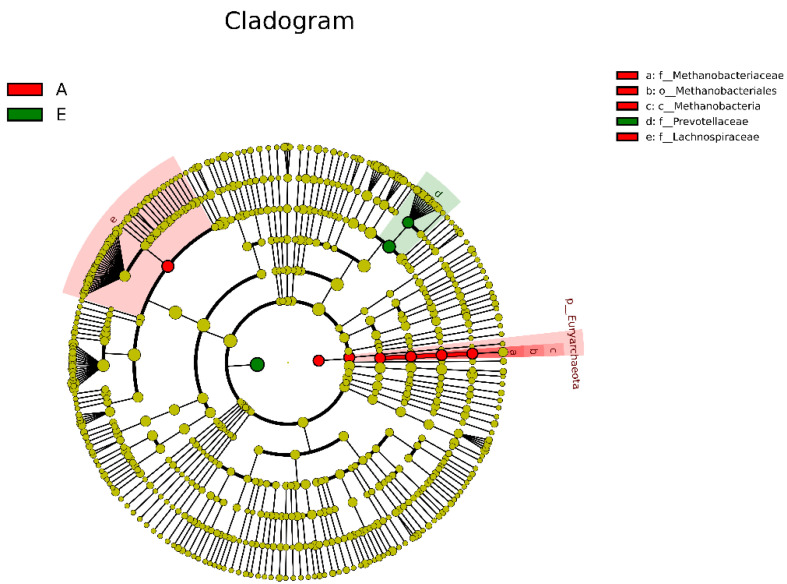
Cladogram of LDA effect size analysis. E, before parturition; A, after parturition. Each small circle represents a species at taxonomic levels (phylum, class, order, family, genus and species, represented by large circles from inside to outside in turn). Yellow circles show species with no significant difference abundance between groups; red (or green) circles show the species with significantly higher abundance in the red group (or green group).

**Figure 5 animals-13-00782-f005:**
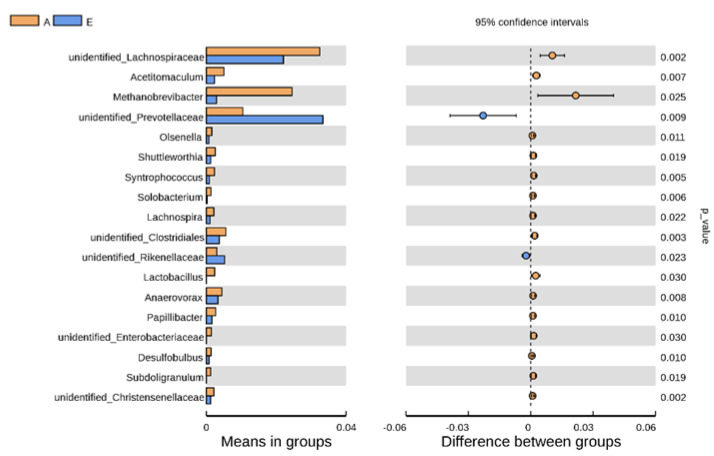
Plot of the *t* test for the relative abundance of the top 30 genera. The left section shows the relative abundance of differential sections between Groups E and A, and each bar represents the mean in the corresponding group of the genera with significant abundance differences between groups. The right section shows the confidence degree of the difference between the groups, and the color in the circle is the same as the color of the group with a high mean.

**Table 1 animals-13-00782-t001:** Regression equations, concentrations, and statistical parameters of each SCFA in the rumen fluid.

Component	Retention Time (min)	Regression Equation	Concentrations (Mean ± SD) (μg/mL)		Statistical Parameters (Before Parturition vs. After Parturition)		
			Before Parturition (*n* = 10)	After Parturition (*n* = 10)	*p* Value	FC	VIP
Acetic acid	3.36	y = 0.64x + 0.03	9575 ± 117.54	6340 ± 294.36	0.01	1.51 ↓	1.21
Propionic acid	3.97	y = 0.46x + 0.001	6109 ± 72.89	3236.8 ± 165.61	0.0004	1.89 ↓	1.36
Isobutyric acid	4.14	y = 1.00x − 0.001	339.4 ± 8.47	366.4 ± 23.23	0.75	0.93	1.18
Butyric acid	4.55	y = 5.57x + 0.20	5218 ± 75.27	3281.7 ± 149.00	0.004	1.59 ↓	1.01
Isovaleric acid	4.8	y = 6.49x − 0.001	345.5 ± 7.17	349.6 ± 20.46	0.96	0.99	1.36
Valeric acid	5.22	y = 7.12x + 0.13	716.7 ± 12.10	416.8 ± 21.69	0.003	1.72	0.63
Caproic acid	5.79	y = 3.39x + 0.23	310.8 ± 8.51	141.7 ± 8.78	0.0006	2.19 ↓	1.15
pH			6.62 ± 0.0624	6.82 ± 0.03	0.0081	0.97 ↑	

Note: ↓ indicates down and ↑ indicates up.

**Table 2 animals-13-00782-t002:** Alpha diversity indices of rumen bacterial communities.

Items	Before Parturition, *n* = 10	After Parturition, *n* = 10	*p* Value
Shannon	7.25 ± 0.48	7.86 ± 0.32	0.0042
Simpson	0.96 ± 0.02	0.97 ± 0.008	0.1762
Chao1	1495.27 ± 115.85	1673.59 ± 157.62	0.0205
ACE	15,224.87 ± 84.09	1701.15 ± 164.32	0.0177

**Table 3 animals-13-00782-t003:** Percentage and difference multiples of the relative abundance of eight major genera before and after parturition (n = 10 cows per group).

Taxonomy	Percentage of the Relative Abundance			SEM	*p* Value	Trend	Difference Multiple
	Before Parturition	After Parturition	□AVE				
Unidentified*_Lachnospiraceae*	2.01	3.99	2.99	0.99	0.002	up	2.00
*Acetitomaculum*	0.23	0.50	0.37	0.14	0.007	up	2.16
*Methanobrevibacter*	0.29	2.46	1.37	1.08	0.025	up	8.43
Unidentified*_Prevotellaceae*	3.34	1.05	2.19	1.14	0.009	down	3.18
*Olsenella*	0.08	0.16	0.12	0.04	0.011	up	2.08
*Syntrophococcus*	0.09	0.23	0.16	0.07	0.005	up	2.63
*Lachnospira*	0.11	0.22	0.16	0.06	0.022	up	2.03
*Lactobacillus*	0.0004	0.24	0.12	0.11	0.030	up	600.25

**Table 4 animals-13-00782-t004:** Spearman Correlation coefficients and *p* values of rumen bacterial genera and SCFAs.

SCFAs	Bacterial Genera	Correlation Coefficient (r)	*p*-Value	AdjPvalue
Acetic acid	*Methanobrevibacter*	−0.601	0.005 **	0.013
	*Acetitomaculum*	−0.469	0.037 *	0.037
	*Syntrophococcus*	−0.472	0.036 *	0.036
	*Lactobacillus*	−0.763	0.000 **	0.000
	Unidentified_*Enterobacteriaceae*	−0.680	0.001 **	0.001
Propionic acid	Unidentified_*Prevotellaceae*	0.477	0.035 *	0.144
	Unidentified_*Lachnospiraceae*	−0.567	0.010 *	0.052
	*Lachnospira*	−0.447	0.048 *	0.241
	*Lactobacillus*	−0.807	0.000 **	0.000
	*Olsenella*	−0.629	0.003 **	0.008
	*Desulfobulbus*	−0.515	0.020 *	0.020
	Unidentified_*Enterobacteriaceae*	−0.730	0.000 **	0.001
Butyric acid	*Acetitomaculum*	−0.476	0.034 *	0.037
	*Syntrophococcus*	−0.489	0.029 *	0.036
	*Lactobacillus*	−0.746	0.000 **	0.000
	*Desulfobulbus*	−0.642	0.002 **	0.006
	*Solobacterium*	−0.716	0.000 **	0.000
	Unidentified_*Enterobacteriaceae*	−0.673	0.001 **	0.001
	*Subdoligranulum*	−0.786	0.000 **	0.000

Note: * indicates *p* < 0.05, ** indicates *p* < 0.01. |r| > 0.7 means a very tight correlation is very close, |r| between 0.4 and 0.7 means a tight correlation.

## Data Availability

None of the data were deposited in an official repository. Data are available upon request.

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
