# Peer review of "Influence of Parturition on Rumen Bacteria and SCFAs in Holstein Cows Based on 16S rRNA Sequencing and Targeted Metabolomics"

_animals, 2023, doi:10.3390/ani13050782_

Round 1

Reviewer 1 Report

Highlighted text must be revised for English or fonts size

1. What is the main question addressed by the research? The objective of the study was to evaluate the effects of parturition on composition and quantity of bacterial communities and concentrations of short-chain fatty acids in dairy cows. 2. Do you consider the topic original or relevant in the field? Yes it is. Does it address a specific gap in the field? Yes, it improves the current knowledge in the field. 3. What does it add to the subject area compared with other published material? I haven't seen something similar to this research. 4. What specific improvements should the authors consider regarding the methodology? What further controls should be considered? No comments. 5. Are the conclusions consistent with the evidence and arguments presented and do they address the main question posed? Yes the conclusions are consistent with the evidence and arguments presented and addressed to main question posed. 6. Are the references appropriate? Yes, they are. 7. Please include any additional comments on the tables and figures. Center contents in table 1.

Author Response

Point 1: Some grammatical errors and improper font should be midified.

Response 1: Thank you again for your positive comments and valuable suggestions to improve the quality of our manuscript. According to your suggestions, we have modified all grammatical errors and improper font. The changes were marked in red in the revised paper and listed below:

“Rumination” is midified as “rumination”. Please see line 38.

Behind “particles” , add “consumed”. Please see line 43.

“Attempting” is midified as “attempting”. Please see line 62.

“in” is midified as”from”. Please see line 92.

“was” is midified as”were”. Please see line 180.

“Two” is midified as”The two”. Please see line 245.

“However” is midified as” however”. Please see line 245.

Behind “abundance”, add “is”. Please see line 285.

Delete “and”, Please see line 303.

   Some improper font is revised and marked up using the “Track Changes” function in revised paper.

   We have centered contents in table 1 and marked up using the “Track Changes” function in revised paper.

Point 2  “Consequently, we adopted 16S rRNA high-throughput  sequencing and targeted GC–MS/MS metabolomics to reveal the influence of parturition on rumen bacterial communities and SCFAs in dairy cows, Attempting to provide more information on physiological process of parturition in Holstein dairy cows” - this part has to do with materials and methods

Response 2: we have modified this part as “Consequently, we collected the rumen fluids from ten cows at Day 3~5 before calving and Day 0 after calving to reveal the influence of parturition on rumen bacterial communities and SCFAs using 16S rRNA high-throughput sequencing and targeted GC–MS/MS metabolomics, attempting to provide more information on physiological process of parturition in Holstein dairy cows”. Please see line 57~63 in revised paper. We hope that the correction will meet with approval.

Reviewer 2 Report

The topic is certainly of considerable interest but the commentary on the results is very generalist. the timing of sampling only limited to only two points, however could be cue to consider effects of these changes in future lactation.
I suggest better describing the methodology related to the sample collection stage and better discussing the results as a function also of productive performance in the lactation stage if they were collected. The discussion is very narrow and limited to reporting similar work.

1. What is the main question addressed by the research? The object of this work was to evaluate the effect of parturition on ruminal microbiota composition

2. Do you consider the topic original or relevant in the field? Does it
address a specific gap in the field? Yes, I consider the research relevant from the point of view of topic and investigation
3. What does it add to the subject area compared with other published
material?
This article enriches the case history concerning the composition of the ruminal microbiota relative to a particular physiological phase.
4. What specific improvements should the authors consider regarding the
methodology? What further controls should be considered? The sample analyzed by consdering the sample type is sufficient; however, there are only two sampling times, which may limit the understanding of the results. The results obtained and their discussion were only briefly contextualized relative to the conditions and available literature results.
5. Are the conclusions consistent with the evidence and arguments presented
and do they address the main question posed? The conclusions are risky since they can be considered relative only to the sample analyzed, and there is no information regarding the influence of the results and changes obtained on the performance of milk production and feed intake after farrowing time.
6. Are the references appropriate? The bibliography is correct
7. Please include any additional comments on the tables and figures.

Author Response

Point 1: I suggest better describing the methodology related to the sample collection stage and better discussing the results as a function also of productive performance in the lactation stage if they were collected. The discussion is very narrow and limited to reporting similar work.

Response 1: Thanks for your constructive suggestion. Accoring to your and other reviewer’s opinion, we have made some modifications in section of sample collection. Please see line 76-78.

    In addition, although we did not coellcet data of productive performance in the lactation stage,  we have discussed the possible effects of SCFAs changes at parturition on performance of milk production and metabolic and inflammatory diseases according to the related literature (Please see line 252-268). We hope that the correction will meet with approval.

Point 2: The sample analyzed by consdering the sample type is sufficient; however, there are only two sampling times, which may limit the understanding of the results. The results obtained and their discussion were only briefly contextualized relative to the conditions and available literature results.

Response 2: Thank you for your suggestion. We agree that this is an important consideration, but we initially designed these two sampling times to better understand the effects of parturition on rumen microorganisms and short-chain fatty acids in dairy cows. If adding sampling times in the postpartum, it may be beyond the scope of this manuscript because ruminal microbiota composition are easily influenced by various factors such as feed, diseases and management.

Point 3: The conclusions are risky since they can be considered relative only to the sample analyzed, and there is no information regarding the influence of the results and changes obtained on the performance of milk production and feed intake after farrowing time.

Response 3: We greatly appreciate this suggestion, more studies would be useful to understand the the influence of parturition on postpatum performance of milk production and feed intake. In the future, we hope to collect more rumen samples and information regarding performance of milk production and feed intake to do correlaction analysis. But we think the experimental design in this paper can provide some valuable informations on physiological process of parturition in Holstein dairy cows. As mentioned in Response 1, we have added some dicussions regarding possible influence of parturition on productive performance and metabolic and inflammatory diseases. We sincerely hope the revised manuscript could be acceptable for you.

Reviewer 3 Report

In this manuscript, the authors investigated and evaluate the effects of parturition on the composition and quantity of bacterial communities and concentrations of short-chain fatty acids. The authors conclude that parturition altered the levels of rumen bacteria and their fermentation ability. I found it interesting, with a good number of animals that can provide statistical significance. However, I have some minor points that I would like to be addressed. And the length is rather short, I would suggest to change the form as short communication if the manuscript could be further accepted.

1. Line 67 “Twenty healthy Holstein cows at expected Day 3~5 before calving (2~3 parity and 2~3.5 BCS) were selected ”while only 10 cows were chosen for sample collection, what is the basis for you to choose these 10 cows?

2. Line114  there exists some spelling error about “2.4.16 .S rRNA"

3. Line 162 In the row“ before parturition ”, please make sure that the number of digits after the decimal point is appropriate

4.  The font size is different in the section of the discussion, please check it.

5.  Line 248267, you referred that decreasing in Prevotellaceae abundance most likely one of the reasons for the 251 “negative energy” balance in postpartum dairy cows, maybe, a correlation analysis between SCFAs and microbiome is better to address this question

Author Response

Point 1: Line 67 “Twenty healthy Holstein cows at expected Day 3~5 before calving (2~3 parity and 2~3.5BCS) were selected ”while only 10 cows were chosen for sample collection, what is the basis for you to choose these 10 cows?

Response 1: We feel great thanks for your professional review work on our article. We selcted twenty healthy Holstein cows according their expected calving date, but the calving date of the ten cows coincided with expected calving, other ten cows did not calve within expected calving date. We have explained the reason in revised paper. Please see line 77.

Point 2: Line114 there exists some spelling error about “2.4.16 .S rRNA".

Response 2: We feel sorry for our carelessness. We have modified as “2.4. 16 S rRNA”, Please see line 118.

Point 3: Line 162 In the row“ before parturition ”, please make sure that the number of digits after the decimal point is appropriate

Response 3: Thank you for pointing this out. We have checked the original data and made sure there were some mistakes as your point in Table 1 . We have modified these mistakes. Please see table 1 in line 168.  

Point 4: The font size is different in the section of the discussion, please check it.

Response 4: We have solved the problem and marked up using the “Track Changes” function in revised paper. Thanks for your reminder.

Point 5: Line 248~267, you referred that decreasing in Prevotellaceae abundance most likely one

of the reasons for the 251 “negative energy” balance in postpartum dairy cows, maybe, a correlation analysis between SCFAs and microbiome is better to address this question.

Response 5: Thanks for your excellent suggestions. We have added the correlation analysis in the revised paper(see line 227-239), and pointed out that propionic acid shows a positive correlationa with Prevotel-laceae in concentration before and after calving in discussion. Please see line 282-283.
